# Hybrid Reward Architecture for Reinforcement Learning

**Harm van Seijen**[1]
harm.vanseijen@microsoft.com

**Mehdi Fatemi**[1]
mehdi.fatemi@microsoft.com

**Joshua Romoff**[1,2]
joshua.romoff@mail.mcgill.ca

**Romain Laroche**[1]
romain.laroche@microsoft.com

**Tavian Barnes**[1]
tavian.barnes@microsoft.com

**Jeffrey Tsang**[1]
tsang.jeffrey@microsoft.com

[1]Microsoft Maluuba, Montreal, Canada
[2]McGill University, Montreal, Canada

## Abstract

One of the main challenges in reinforcement learning (RL) is generalisation. In typical deep RL methods this is achieved by approximating the optimal value function with a low-dimensional representation using a deep network. While this approach works well in many domains, in domains where the optimal value function cannot easily be reduced to a low-dimensional representation, learning can be very slow and unstable. This paper contributes towards tackling such challenging domains, by proposing a new method, called Hybrid Reward Architecture (HRA). HRA takes as input a decomposed reward function and learns a separate value function for each component reward function. Because each component typically only depends on a subset of all features, the corresponding value function can be approximated more easily by a low-dimensional representation, enabling more effective learning. We demonstrate HRA on a toy-problem and the Atari game Ms. Pac-Man, where HRA achieves above-human performance.

## 1   Introduction

In reinforcement learning (RL) (Sutton & Barto, 1998; Szepesvári, 2009), the goal is to find a behaviour policy that maximises the return—the discounted sum of rewards received over time—in a data-driven way. One of the main challenges of RL is to scale methods such that they can be applied to large, real-world problems. Because the state-space of such problems is typically massive, strong generalisation is required to learn a good policy efficiently.

Mnih et al. (2015) achieved a big breakthrough in this area: by combining standard RL techniques with deep neural networks, they achieved above-human performance on a large number of Atari 2600 games, by learning a policy from pixels. The generalisation properties of their Deep $Q$-Networks (DQN) method is achieved by approximating the optimal value function. A value function plays an important role in RL, because it predicts the expected return, conditioned on a state or state-action pair. Once the optimal value function is known, an optimal policy can be derived by acting greedily with respect to it. By modelling the current estimate of the optimal value function with a deep neural network, DQN carries out a strong generalisation on the value function, and hence on the policy.

The generalisation behaviour of DQN is achieved by regularisation on the model for the optimal value function. However, if the optimal value function is very complex, then learning an accurate low-dimensional representation can be challenging or even impossible. Therefore, when the optimal value function cannot easily be reduced to a low-dimensional representation, we argue to apply a complementary form of regularisation on the target side. Specifically, we propose to replace the optimal value function as target for training with an alternative value function that is easier to learn, but still yields a reasonable—but generally not optimal—policy, when acting greedily with respect to it.

The key observation behind regularisation on the target function is that two very different value functions can result in the same policy when an agent acts greedily with respect to them. At the same time, some value functions are much easier to learn than others. Intrinsic motivation (Stout et al., 2005; Schmidhuber, 2010) uses this observation to improve learning in sparse-reward domains, by adding a domain-specific intrinsic reward signal to the reward coming from the environment. When the intrinsic reward function is potential-based, optimality of the resulting policy is maintained (Ng et al., 1999). In our case, we aim for simpler value functions that are easier to represent with a low-dimensional representation.

Our main strategy for constructing an easy-to-learn value function is to decompose the reward function of the environment into $n$ different reward functions. Each of them is assigned a separate reinforcement-learning agent. Similar to the Horde architecture (Sutton et al., 2011), all these agents can learn in parallel on the same sample sequence by using off-policy learning. Each agent gives its action-values of the current state to an aggregator, which combines them into a single value for each action. The current action is selected based on these aggregated values.

We test our approach on two domains: a toy-problem, where an agent has to eat 5 randomly located fruits, and Ms. Pac-Man, one of the hard games from the ALE benchmark set (Bellemare et al., 2013).

## 2 Related Work

Our HRA method builds upon the Horde architecture (Sutton et al., 2011). The Horde architecture consists of a large number of 'demons' that learn in parallel via off-policy learning. Each demon trains a separate general value function (GVF) based on its own policy and *pseudo-reward* function. A pseudo-reward can be any feature-based signal that encodes useful information. The Horde architecture is focused on building up general knowledge about the world, encoded via a large number of GVFs. HRA focusses on training separate components of the environment-reward function, in order to more efficiently learn a control policy. UVFA (Schaul et al., 2015) builds on Horde as well, but extends it along a different direction. UVFA enables generalization across different tasks/goals. It does not address how to solve a single, complex task, which is the focus of HRA.

Learning with respect to multiple reward functions is also a topic of multi-objective learning (Roijers et al., 2013). So alternatively, HRA can be viewed as applying multi-objective learning in order to more efficiently learn a policy for a single reward function.

Reward function decomposition has been studied among others by Russell & Zimdar (2003) and Sprague & Ballard (2003). This earlier work focusses on strategies that achieve optimal behavior. Our work is aimed at improving learning-efficiency by using simpler value functions and relaxing optimality requirements.

There are also similarities between HRA and UNREAL (Jaderberg et al., 2017). Notably, both solve multiple smaller problems in order to tackle one hard problem. However, the two architectures are different in their workings, as well as the type of challenge they address. UNREAL is a technique that boosts representation learning in difficult scenarios. It does so by using auxiliary tasks to help train the lower-level layers of a deep neural network. An example of such a challenging representation-learning scenario is learning to navigate in the 3D Labyrinth domain. On Atari games, the reported performance gain of UNREAL is minimal, suggesting that the standard deep RL architecture is sufficiently powerful to extract the relevant representation. By contrast, the HRA architecture breaks down a task into smaller pieces. HRA's multiple smaller tasks are not unsupervised; they are tasks that are directly relevant to the main task. Furthermore, whereas UNREAL is inherently a deep RL technique, HRA is agnostic to the type of function approximation used. It can be combined with deep

neural networks, but it also works with exact, tabular representations. HRA is useful for domains where having a high-quality representation is not sufficient to solve the task efficiently.

Diuk's object-oriented approach (Diuk et al., 2008) was one of the first methods to show efficient learning in video games. This approach exploits domain knowledge related to the transition dynamic to efficiently learn a compact transition model, which can then be used to find a solution using dynamic-programming techniques. This inherently model-based approach has the drawback that while it efficiently learns a very compact model of the transition dynamics, it does not reduce the state-space of the problem. Hence, it does not address the main challenge of Ms. Pac-Man: its huge state-space, which is even for DP methods intractable (Diuk applied his method to an Atari game with only 6 objects, whereas Ms. Pac-Man has over 150 objects).

Finally, HRA relates to options (Sutton et al., 1999; Bacon et al., 2017), and more generally hierarchical learning (Barto & Mahadevan, 2003; Kulkarni et al., 2016). Options are temporally-extended actions that, like HRA's heads, can be trained in parallel based on their own (intrinsic) reward functions. However, once an option has been trained, the role of its intrinsic reward function is over. A higher-level agent that uses an option sees it as just another action and evaluates it using its own reward function. This can yield great speed-ups in learning and help substantially with better exploration, but they do not directly make the value function of the higher-level agent less complex. The heads of HRA represent values, trained with components of the environment reward. Even after training, these values stay relevant, because the aggregator uses them to select its action.

## 3 Model

Consider a Markov Decision Process $\langle \mathcal{S}, \mathcal{A}, P, R_{env}, \gamma \rangle$, which models an agent interacting with an environment at discrete time steps $t$. It has a state set $\mathcal{S}$, action set $\mathcal{A}$, environment reward function $R_{env} : \mathcal{S} \times \mathcal{A} \times \mathcal{S} \to \mathbb{R}$, and transition probability function $P : \mathcal{S} \times \mathcal{A} \times \mathcal{S} \to [0, 1]$. At time step $t$, the agent observes state $s_t \in \mathcal{S}$ and takes action $a_t \in \mathcal{A}$. The agent observes the next state $s_{t+1}$, drawn from the transition probability distribution $P(s_t, a_t, \cdot)$, and a reward $r_t = R_{env}(s_t, a_t, s_{t+1})$. The behaviour is defined by a policy $\pi : \mathcal{S} \times \mathcal{A} \to [0, 1]$, which represents the selection probabilities over actions. The goal of an agent is to find a policy that maximises the expectation of the return, which is the discounted sum of rewards: $G_t := \sum_{i=0}^{\infty} \gamma^i r_{t+i}$, where the discount factor $\gamma \in [0, 1]$ controls the importance of immediate rewards versus future rewards. Each policy $\pi$ has a corresponding action-value function that gives the expected return conditioned on the state and action, when acting according to that policy:

$$Q^{\pi}(s, a) = \mathbb{E}[G_t | s_t = s, a_t = a, \pi] \tag{1}$$

The optimal policy $\pi^*$ can be found by iteratively improving an estimate of the optimal action-value function $Q^*(s, a) := \max_{\pi} Q^{\pi}(s, a)$, using sample-based updates. Once $Q^*$ is sufficiently accurate approximated, acting greedy with respect to it yields the optimal policy.

### 3.1 Hybrid Reward Architecture

The $Q$-value function is commonly estimated using a function approximator with weight vector $\theta$: $Q(s, a; \theta)$. DQN uses a deep neural network as function approximator and iteratively improves an estimate of $Q^*$ by minimising the sequence of loss functions:

$$\mathcal{L}_i(\theta_i) = \mathbb{E}_{s,a,r,s'}[(y_i^{DQN} - Q(s, a; \theta_i))^2], \tag{2}$$

$$\text{with} \quad y_i^{DQN} = r + \gamma \max_{a'} Q(s', a'; \theta_{i-1}), \tag{3}$$

The weight vector from the previous iteration, $\theta_{i-1}$, is encoded using a separate target network.

We refer to the $Q$-value function that minimises the loss function(s) as the *training target*. We will call a training target *consistent*, if acting greedily with respect to it results in a policy that is optimal under the reward function of the environment; we call a training target *semi-consistent*, if acting greedily with respect to it results in a good policy—but not an optimal one—under the reward function of the environment. For (2), the training target is $Q_{env}^*$, the optimal action-value function under $R_{env}$, which is the default consistent training target.

That a training target is consistent says nothing about how easy it is to learn that target. For example, if $R_{env}$ is sparse, the default learning objective can be very hard to learn. In this case, adding a

potential-based additional reward signal to $R_{env}$ can yield an alternative consistent learning objective that is easier to learn. But a sparse environment reward is not the only reason a training target can be hard to learn. We aim to find an alternative training target for domains where the default training target $Q_{env}^*$ is hard to learn, due to the function being high-dimensional and hard to generalise for. Our approach is based on a decomposition of the reward function.

We propose to decompose the reward function $R_{env}$ into $n$ reward functions:

$$R_{env}(s, a, s') = \sum_{k=1}^{n} R_k(s, a, s'), \qquad \text{for all } s, a, s', \tag{4}$$

and to train a separate reinforcement-learning agent on each of these reward functions. There are infinitely many different decompositions of a reward function possible, but to achieve value functions that are easy to learn, the decomposition should be such that each reward function is mainly affected by only a small number of state variables.

Because each agent $k$ has its own reward function, it has also its own $Q$-value function, $Q_k$. In general, different agents can share multiple lower-level layers of a deep $Q$-network. Hence, we will use a single vector $\theta$ to describe the combined weights of the agents. We refer to the combined network that represents all $Q$-value functions as the Hybrid Reward Architecture (HRA) (see Figure 1). Action selection for HRA is based on the sum of the agent's $Q$-value functions, which we call $Q_{\text{HRA}}$:

$$Q_{\text{HRA}}(s, a; \theta) := \sum_{k=1}^{n} Q_k(s, a; \theta), \qquad \text{for all } s, a. \tag{5}$$

The collection of agents can be viewed alternatively as a single agent with multiple *heads*, with each head producing the action-values of the current state under a different reward function.

The sequence of loss function associated with HRA is:

$$\mathcal{L}_i(\theta_i) = \mathbb{E}_{s,a,r,s'} \left[ \sum_{k=1}^{n} (y_{k,i} - Q_k(s, a; \theta_i))^2 \right], \tag{6}$$

$$\text{with} \qquad y_{k,i} = R_k(s, a, s') + \gamma \max_{a'} Q_k(s', a'; \theta_{i-1}). \tag{7}$$

By minimising these loss functions, the different heads of HRA approximate the optimal action-value functions under the different reward functions: $Q_1^*, \ldots, Q_n^*$. Furthermore, $Q_{\text{HRA}}$ approximates $Q_{\text{HRA}}^*$, defined as:

$$Q_{\text{HRA}}^*(s, a) := \sum_{k=1}^{n} Q_k^*(s, a) \qquad \text{for all } s, a.$$

Note that $Q_{\text{HRA}}^*$ is different from $Q_{env}^*$ and generally not consistent.

An alternative training target is one that results from evaluating the uniformly random policy $\upsilon$ under each component reward function: $Q_{\text{HRA}}^\upsilon(s, a) := \sum_{k=1}^{n} Q_k^\upsilon(s, a)$. $Q_{\text{HRA}}^\upsilon$ is equal to $Q_{env}^\upsilon$, the

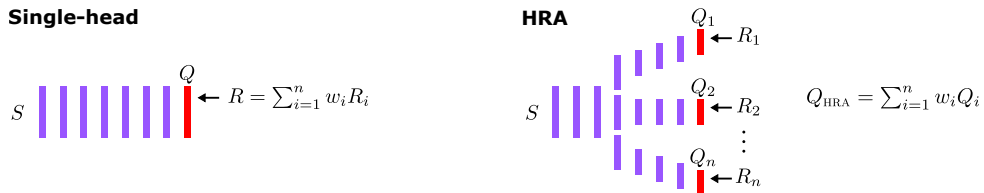

Figure 1: Illustration of Hybrid Reward Architecture.

$Q$-values of the random policy under $R_{env}$, as shown below:

$$Q_{env}^v(s,a) = \mathbb{E}\left[\sum_{i=0}^{\infty}\gamma^i R_{env}(s_{t+i},a_{t+i},s_{t+1+i})|s_t=s,a_t=a,v\right],$$

$$= \mathbb{E}\left[\sum_{i=0}^{\infty}\gamma^i\sum_{k=1}^{n}R_k(s_{t+i},a_{t+i},s_{t+1+i})|s_t=s,a_t=a,v\right],$$

$$= \sum_{k=1}^{n}\mathbb{E}\left[\sum_{i=0}^{\infty}\gamma^i R_k(s_{t+i},a_{t+i},s_{t+1+i})|s_t=s,a_t=a,v\right],$$

$$= \sum_{k=1}^{n}Q_k^v(s,a) := Q_{\text{HRA}}^v(s,a).$$

This training target can be learned using the expected Sarsa update rule (van Seijen et al., 2009), by replacing (7), with

$$y_{k,i} = R_k(s,a,s') + \gamma\sum_{a'\in\mathcal{A}}\frac{1}{|\mathcal{A}|}Q_k(s',a';\theta_{i-1}). \tag{8}$$

Acting greedily with respect to the Q-values of a random policy might appear to yield a policy that is just slightly better than random, but, surpringly, we found that for many navigation-based domains $Q_{\text{HRA}}^v$ acts as a semi-consistent training target.

### 3.2 Improving Performance further by using high-level domain knowledge.

In its basic setting, the only domain knowledge applied to HRA is in the form of the decomposed reward function. However, one of the strengths of HRA is that it can easily exploit more domain knowledge, if available. Domain knowledge can be exploited in one of the following ways:

1. **Removing irrelevant features.** Features that do not affect the received reward in any way (directly or indirectly) only add noise to the learning process and can be removed.

2. **Identifying terminal states.** Terminal states are states from which no further reward can be received; they have by definition a value of 0. Using this knowledge, HRA can refrain from approximating this value by the value network, such that the weights can be fully used to represent the non-terminal states.

3. **Using pseudo-reward functions.** Instead of updating a head of HRA using a component of the environment reward, it can be updated using a pseudo-reward. In this scenario, a set of GVFs is trained in parallel using pseudo-rewards.

While these approaches are not specific to HRA, HRA can exploit domain knowledge to a much great extend, because it can apply these approaches to each head individually. We show this empirically in Section 4.1.

## 4 Experiments

### 4.1 Fruit Collection task

In our first domain, we consider an agent that has to collect fruits as quickly as possible in a $10 \times 10$ grid. There are 10 possible fruit locations, spread out across the grid. For each episode, a fruit is randomly placed on 5 of those 10 locations. The agent starts at a random position. The reward is +1 if a fruit gets eaten and 0 otherwise. An episode ends after all 5 fruits have been eaten or after 300 steps, whichever comes first.

We compare the performance of DQN with HRA using the same network. For HRA, we decompose the reward function into 10 different reward functions, one per possible fruit location. The network consists of a binary input layer of length 110, encoding the agent's position and whether there is a fruit on each location. This is followed by a fully connected hidden layer of length 250. This layer is connected to 10 heads consisting of 4 linear nodes each, representing the action-values of

the 4 actions under the different reward functions. Finally, the mean of all nodes across heads is computed using a final linear layer of length 4 that connects the output of corresponding nodes in each head. This layer has fixed weights with value 1 (i.e., it implements Equation 5). The difference between HRA and DQN is that DQN updates the network from the fourth layer using loss function (2), whereas HRA updates the network from the third layer using loss function (6).

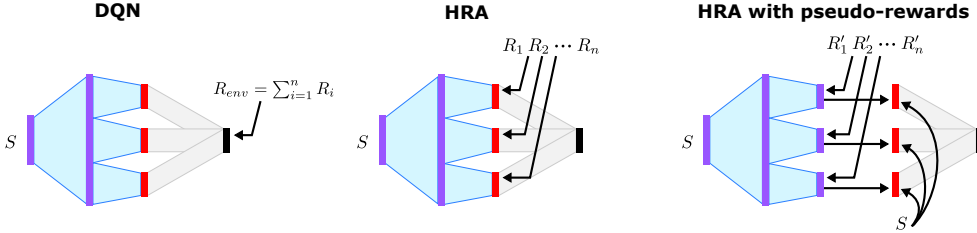

Figure 2: The different network architectures used.

Besides the full network, we test using different levels of domain knowledge, as outlined in Section 3.2: 1) removing the irrelevant features for each head (providing only the position of the agent + the corresponding fruit feature); 2) the above plus identifying terminal states; 3) the above plus using pseudo rewards for learning GVFs to go to each of the 10 locations (instead of learning a value function associated to the fruit at each location). The advantage is that these GVFs can be trained even if there is no fruit at a location. The head for a particular location copies the $Q$-values of the corresponding GVF if the location currently contains a fruit, or outputs 0s otherwise. We refer to these as *HRA+1*, *HRA+2* and *HRA+3*, respectively. For DQN, we also tested a version that was applied to the same network as *HRA+1*; we refer to this version as *DQN+1*.

Training samples are generated by a random policy; the training process is tracked by evaluating the greedy policy with respect to the learned value function after every episode. For HRA, we performed experiments with $Q^*_{\text{HRA}}$ as training target (using Equation 7), as well as $Q^v_{\text{HRA}}$ (using Equation 8). Similarly, for DQN we used the default training target, $Q^*_{env}$, as well as $Q^v_{env}$. We optimised the step-size and the discount factor for each method separately.

The results are shown in Figure 3 for the best settings of each method. For DQN, using $Q^*_{env}$ as training target resulted in the best performance, while for HRA, using $Q^v_{\text{HRA}}$ resulted in the best performance. Overall, HRA shows a clear performance boost over DQN, even though the network is identical. Furthermore, adding different forms of domain knowledge causes further large improvements. Whereas using a network structure enhanced by domain knowledge improves performance of HRA, using that same network for DQN results in a decrease in performance. The big boost in performance that occurs when the the terminal states are identified is due to the representation becoming a one-hot vector. Hence, we removed the hidden layer and directly fed this one-hot vector

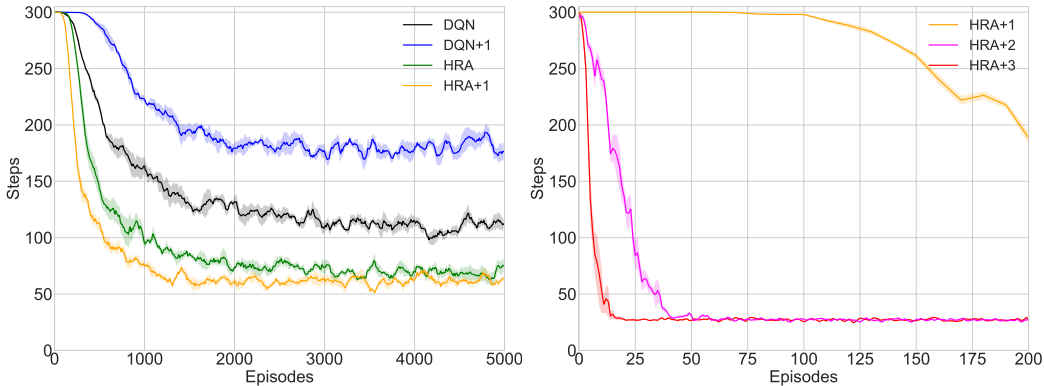

Figure 3: Results on the fruit collection domain, in which an agent has to eat 5 randomly placed fruits. An episode ends after all 5 fruits are eaten or after 300 steps, whichever comes first.

into the different heads. Because the heads are linear, this representation reduces to an exact, tabular representation. For the tabular representation, we used the same step-size as the optimal step-size for the deep network version.

## 4.2 ATARI game: Ms. Pac-Man

Our second domain is the Atari 2600 game Ms. Pac-Man (see Figure 4). Points are obtained by eating pellets, while avoiding ghosts (contact with one causes Ms. Pac-Man to lose a life). Eating one of the special power pellets turns the ghosts blue for a small duration, allowing them to be eaten for extra points. Bonus fruits can be eaten for further points, twice per level. When all pellets have been eaten, a new level is started. There are a total of 4 different maps and 7 different fruit types, each with a different point value. We provide full details on the domain in the supplementary material.

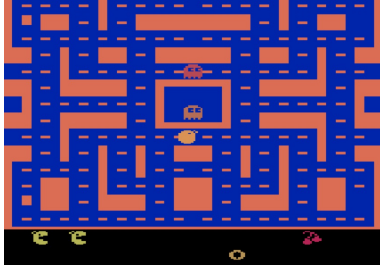

Figure 4: The game Ms. Pac-Man.

**Baselines.** While our version of Ms. Pac-Man is the same as used in literature, we use different preprocessing. Hence, to test the effect of our preprocessing, we implement the A3C method (Mnih et al., 2016) and run it with our preprocessing. We refer to the version with our preprocessing as 'A3C(channels)', the version with the standard preprocessing 'A3C(pixels)', and A3C's score reported in literature 'A3C(reported)'.

**Preprocessing.** Each frame from ALE is $210 \times 160$ pixels. We cut the bottom part and the top part of the screen to end up with $160 \times 160$ pixels. From this, we extract the position of the different objects and create for each object a separate input channel, encoding its location with an accuracy of 4 pixels. This results in 11 binary channels of size $40 \times 40$. Specifically, there is a channel for Ms. Pac-Man, each of the four ghosts, each of the four blue ghosts (these are treated as different objects), the fruit plus one channel with all the pellets (including power pellets). For A3C, we combine the 4 channels of the ghosts into a single channel, to allow it to generalise better across ghosts. We do the same with the 4 channels of the blue ghosts. Instead of giving the history of the last 4 frames as done in literature, we give the orientation of Ms. Pac-Man as a 1-hot vector of length 4 (representing the 4 compass directions).

**HRA architecture.** The environment reward signal corresponds with the points scored in the game. Before decomposing the reward function, we perform reward shaping by adding a negative reward of -1000 for contact with a ghost (which causes Ms. Pac-Man to lose a life). After this, the reward is decomposed in a way that each object in the game (pellet/fruit/ghost/blue ghost) has its own reward function. Hence, there is a separate RL agent associated with each object in the game that estimates a Q-value function of its corresponding reward function.

To estimate each component reward function, we use the three forms of domain knowledge discussed in Section 3.2. HRA uses GVFs that learn pseudo Q-values (with values in the range [0, 1]) for getting to a particular location on the map (separate GVFs are learnt for each of the four maps). In contrast to the fruit collection task (Section 4.1), HRA learns part of its representation during training: it starts off with 0 GVFs and 0 heads for the pellets. By wandering around the maze, it discovers new map locations it can reach, resulting in new GVFs being created. Whenever the agent finds a pellet at a new location it creates a new head corresponding to the pellet.

The $Q$-values for an object (pellet/fruit/ghost/blue ghost) are set to the pseudo $Q$-values of the GVF corresponding with the object's location (i.e., moving objects use a different GVF each time), multiplied with a weight that is set equal to the reward received when the object is eaten. If an object is not on the screen, all its $Q$-values are 0.

We test two aggregator types. The first one is a linear one that sums the $Q$-values of all heads (see Equation 5). For the second one, we take the sum of all the heads that produce points, and normalise the resulting $Q$-values; then, we add the sum of the $Q$-values of the heads of the regular ghosts, multiplied with a weight vector.

For exploration, we test two complementary types of exploration. Each type adds an extra exploration head to the architecture. The first type, which we call *diversification*, produces random $Q$-values, drawn from a uniform distribution over [0, 20]. We find that it is only necessary during the first 50 steps, to ensure starting each episode randomly. The second type, which we call *count-based*, adds a bonus for state-action pairs that have not been explored a lot. It is inspired by upper confidence bounds (Auer et al., 2002). Full details can be found in the supplementary material.

For our final experiment, we implement a special head inspired by *executive-memory* literature (Fuster, 2003; Gluck et al., 2013). When a human game player reaches the maximum of his cognitive and physical ability, he starts to look for favourable situations or even glitches and memorises them. This cognitive process is indeed memorising a sequence of actions (also called habit), and is not necessarily optimal. Our executive-memory head records every sequence of actions that led to pass a level without any kill. Then, when facing the same level, the head gives a very high value to the recorded action, in order to force the aggregator's selection. Note that our simplified version of executive memory does not generalise.

**Evaluation metrics.** There are two different evaluation methods used across literature which result in very different scores. Because ALE is ultimately a deterministic environment (it implements pseudo-randomness using a random number generator that always starts with the same seed), both evaluation metrics aim to create randomness in the evaluation in order to rate methods with more generalising behaviour higher. The first metric introduces a mild form of randomness by taking a random number of no-op actions before control is handed over to the learning algorithm. In the case of Ms. Pac-Man, however, the game starts with a certain inactive period that exceeds the maximum number of no-op steps, resulting in the game having a fixed start after all. The second metric selects random starting points along a human trajectory and results in much stronger randomness, and does result in the intended random start evaluation. We refer to these metrics as 'fixed start' and 'random start'.

**Results.** Figure 5 shows the training curves; Table 1 shows the final score after training. The best reported fixed start score comes from STRAW (Vezhnevets et al., 2016); the best reported random start score comes from the Dueling network architecture (Wang et al., 2016). The human fixed start score comes from Mnih et al. (2015); the human random start score comes from Nair et al. (2015). We train A3C for 800 million frames. Because HRA learns fast, we train it only for 5,000 episodes, corresponding with about 150 million frames (note that better policies

Table 1: Final scores.

| method | fixed start | random start |
|---|---|---|
| best reported | 6,673 | 2,251 |
| human | 15,693 | 15,375 |
| A3C (reported) | — | 654 |
| A3C (pixels) | 2,168 | 626 |
| A3C (channels) | 2,423 | 589 |
| **HRA** | **25,304** | **23,770** |

result in more frames per episode). We tried a few different settings for HRA: with/without normalisation and with/without each type of exploration. The score shown for HRA uses the best combination: with normalisation and with both exploration types. All combinations achieved over 10,000 points in training, except the combination with no exploration at all, which—not surprisingly—performed very poorly. With the best combination, HRA not only outperforms the state-of-the-art on both metrics, it also significantly outperforms the human score, convincingly demonstrating the strength of HRA.

Comparing A3C(pixels) and A3C(channels) in Table 1 reveals a surprising result: while we use advanced preprocessing by separating the screen image into relevant object channels, this did not significantly change the performance of A3C.

In our final experiment, we test how well HRA does if it exploits the weakness of the fixed-start evaluation metric by using a simplified version of executive memory. Using this version, we not only surpass the human high-score of 266,330 points,[1] we achieve the maximum possible score of 999,990 points in less than 3,000 episodes. The curve is slow in the first stages because the model has to be trained, but even though the further levels get more and more difficult, the level passing speeds up by taking advantage of already knowing the maps. Obtaining more points is impossible, not because the game ends, but because the score overflows to 0 when reaching a million points.[2]

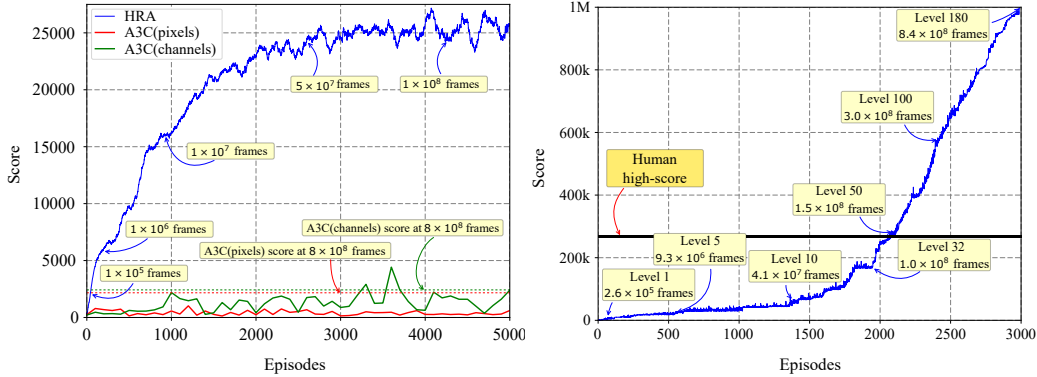

Figure 5: Training smoothed over 100 episodes. Figure 6: Training with trajectory memorisation.

## 5 Discussion

One of the strengths of HRA is that it can exploit domain knowledge to a much greater extent than single-head methods. This is clearly shown by the fruit collection task: while removing irrelevant features improves performance of HRA, the performance of DQN decreased when provided with the same network architecture. Furthermore, separating the pixel image into multiple binary channels only makes a small improvement in the performance of A3C over learning directly from pixels. This demonstrates that the reason that modern deep RL struggle with Ms. Pac-Man is not related to learning from pixels; the underlying issue is that the optimal value function for Ms. Pac-Man cannot easily be mapped to a low-dimensional representation.

HRA solves Ms. Pac-Man by learning close to 1,800 general value functions. This results in an exponential breakdown of the problem size: whereas the input state-space corresponding with the binary channels is in the order of $10^{77}$, each GVF has a state-space in the order of $10^3$ states, small enough to be represented without any function approximation. While we could have used a deep network for representing each GVF, using a deep network for such small problems hurts more than it helps, as evidenced by the experiments on the fruit collection domain.

We argue that many real-world tasks allow for reward decomposition. Even if the reward function can only be decomposed in two or three components, this can already help a lot, due to the exponential decrease of the problem size that decomposition might cause.

## Footnotes

[1]See highscore.com: 'Ms. Pac-Man (Atari 2600 Emulated)'.

[2]For a video of HRA's final trajectory reaching this point, see: https://youtu.be/VeXNw0Owf0Y

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
