[Supplementary Material · supp_material_crc.pdf]

# A  Ms. Pac-Man - experimental details

## A.1  General information about Atari 2600 Ms. Pac-Man

The second domain is the Atari 2600 game Ms. Pac-Man. Points are obtained by eating pellets, while avoiding ghosts (contact with one causes Ms. Pac-Man to lose a life). Eating one of the special power pellets turns the ghost blue for a small duration, allowing them to be eaten for extra points. Bonus fruits can be eaten for further increasing points, twice per level. When all pellets have been eaten, a new level is started. There are a total of 4 different maps (see Figure 1 and Table 1) and 7 different fruit types, each with a different point value (see Table 2).

Ms. Pac-Man is considered as one of hard games from the ALE benchmark set. When comparing performance, it is important to realise that there are two different evaluation methods for ALE games used across literature which result in hugely different scores (see Table 3). Because ALE is ultimately a deterministic environment (it implements pseudo-randomness using a random number generator that always starts with the same seed), both evaluation metrics aim to create randomness in the evaluation in order to discourage methods from exploiting this deterministic property and rate methods with more generalising behaviour higher. The first metric introduces a mild form of randomness by taking a random number of no-op actions before control is handed over to the learning algorithm. In the case of Ms. Pac-Man, however, the game starts with a certain inactive period that exceeds the maximum number of random no-op steps, resulting in the game having a fixed start after all. The second metric selects random starting points along a human trajectory and results in much stronger randomness, and does result in the intended random start evaluation.

The best method with fixed start evaluation is STRAW with 6.673 points (Vezhnevets et al., 2016); the best with random start evaluation is the dueling network architecture with 2.251 points (Wang et al., 2016). The human baseline, as reported by Mnih et al. (2015), is 15.693 points. The highest reported score by a human is 266.330. For reference, A3C scored 654 points with random start evaluation (Mnih et al., 2016); no score is reported for fixed start evaluation.

Figure 1: The four different maps of Ms. Pac-Man.

Table 1: Map type and fruit type per level.

| level | map | fruit |
|---|---|---|
| 1 | red | cherry |
| 2 | red | strawberry |
| 3 | blue | orange |
| 4 | blue | pretzel |
| 5 | white | apple |
| 6 | white | pear |
| 7 | green | banana |
| 8 | green | $\langle random \rangle$ |
| 9 | white | $\langle random \rangle$ |
| 10 | green | $\langle random \rangle$ |
| 11 | white | $\langle random \rangle$ |
| 12 | green | $\langle random \rangle$ |
| ⋮ | ⋮ | ⋮ |

Table 2: Points breakdown of edible objects.

| object | points |
|---|---|
| pellet | 10 |
| power pellet | 50 |
| $1^{st}$ blue ghost | 200 |
| $2^{nd}$ blue ghost | 400 |
| $3^{th}$ blue ghost | 800 |
| $4^{th}$ blue ghost | 1,600 |
| cherry | 100 |
| strawberry | 200 |
| orange | 500 |
| pretzel | 700 |
| apple | 1,000 |
| pear | 2,000 |
| banana | 5,000 |

Table 3: Reported scores on Ms. Pac-Man for fixed start evaluation (called 'random no-ops' in literature) and random start evaluation ('human starts' in literature).

| algorithm | fixed start | source | rand start | source |
|---|---|---|---|---|
| Human | 15,693 | Mnih et al. (2015) | 15,375 | Nair et al. (2015) |
| Random | 308 | Mnih et al. (2015) | 198 | Nair et al. (2015) |
| DQN | 2,311 | Mnih et al. (2015) | 764 | Nair et al. (2015) |
| DDQN | 3,210 | van Hasselt et al. (2016b) | 1,241 | van Hasselt et al. (2016b) |
| Prio. Exp. Rep | 6,519 | Schaul et al. (2016) | 1,825 | Schaul et al. (2016) |
| Dueling | 6,284 | Wang et al. (2016) | 2,251 | Wang et al. (2016) |
| A3C | — | — | 654 | Mnih et al. (2016) |
| Gorila | 3,234 | Nair et al. (2015) | 1,263 | Nair et al. (2015) |
| Pop-Art | 4,964 | van Hasselt et al. (2016a) | — | — |
| STRAW | 6,673 | Vezhnevets et al. (2016) | — | — |
| **HRA** | **25,304** | (this paper) | **23,770** | (this paper) |

## A.2 HRA architecture

**GVF heads.** Ms. Pac-Man state is defined as its position on the map and her direction (heading North, East, South or West). Depending on the map, there are about 400 positions and 950 states (not all directions are possible for each position). A GVF is created online for each visited Ms. Pac-Man position. Each GVF is then in charge of determining the value of the random policy of Ms. Pac-Man state for getting the pseudo-reward placed on the GVF's associated position. The GVFs are trained online with off-policy 1-step bootstrapping with $\alpha = 1$ and $\gamma = 0.99$. Thus, the full tabular representation of the GVF grid contains $nb_{maps} \times nb_{positions} \times nb_{states} \times nb_{actions} \approx 14M$ entries.

**Aggregator.** For each object of the game: pellets, ghosts and fruits, the GVF corresponding to its position is activated with a multiplier depending on the object type. Edible objects' multipliers are consistent with the number of points they grant: pellets' multiplier is 10, power pellets' 50, fruits' 200, and blue and edible ghosts' 1,000. A ghosts' multiplier of -1,000 has demonstrated to be a fair balance between gaining points and not being killed. Finally, the aggregator sums up all the activated and multiplied GVFs to compute a global score for each of the nine actions and chooses the action that maximises it.

**Diversification head.** The blue curve on Figure 2 reveals that this naïve setting performs bad because it tends to deterministically repeat a bad trajectory like a robot hitting a wall continuously. In order to avoid this pitfall, we need to add an exploratory mechanism. An $\epsilon$-greedy exploration is not suitable for this problem since it might unnecessarily put Ms. Pac-Man in danger. A Boltzmann

Figure 2: Training curves for incrementally head additions to the HRA architecture.

distributed exploration is more suitable because it favours exploring the safe actions. It would be possible to apply it on top of the aggregator, but we chose here to add a diversification head $Q_{div}(s_t, a)$ that generates for each action a random value. This random value is drawn according to a uniform distribution in [0,20]. We found that it is only necessary during the 50 first steps, to ensure starting each episode randomly.

$$Q_{div}(s_t, a) \sim \mathcal{U}(0, 20) \qquad \text{if } t < 50, \qquad (1)$$
$$Q_{div}(s_t, a) = 0 \qquad \text{otherwise.} \qquad (2)$$

**Score heads normalisation.** The orange curve on Figure 2 shows that the diversification head solves the determinism issue. The so-built architecture progresses fast, up to 10,000 points, but then starts regressing. The analysis of the generated trajectories reveals that the system has difficulty to finish levels: indeed, when only a few pellets remain on the screen, the aggregator gets overwhelmed by the ghost avoider values. The regression in score is explained by the fact that the more the system learns the more is gets easily scared by the ghosts, and therefore the more difficult it is for it to finish the levels. We solve this issue by modifying the additive aggregator with a normalisation over the score heads between 0 and 1. To fit this new value scale, the ghost multiplier is modified to -10.

**Targeted exploration head.** The green curve on Figure 2 grows over time as expected. It might be surprising at first look that the orange curve grows faster, but it is because the episodes without normalisation tend to last much longer, which allows more GVF updates per episode. In order to speed up the learning, we decide to use a targeted exploration head $Q_{teh}(s, a)$, that is motivated by trying out the less explored state-action couples. The value of this agent is computed as follows:

$$Q_{teh}(s, a) = \kappa \sqrt{\frac{\sqrt[4]{N}}{n(s, a)}}, \qquad (3)$$

where $N$ is the number of actions taken until now and $n(s, a)$ the number of times action $a$ has been performed in state $s$. This formula is inspired from upper confidence bounds Auer et al. (2002), but replacing the stochastically motivated logarithmic function by a less drastic one is more compliant

with our need for bootstrapping propagation. Note that this targeted exploration head is not a replacement for the diversification head. They are complementary: diversification for making each trajectory unique, and targeted exploration for prioritised exploration. The red curve on Figure 2 reveals that the new targeted exploration head helps exploration and makes the learning faster. This setting constitutes the HRA architecture that is used in every experiment.

**Executive memory head.** When a human game player reaches the maximum of his cognitive and physical ability, he starts to look for favourable situations or even glitches and memorises them. This cognitive process is referred as executive memory in cognitive science literature (Fuster, 2003; Gluck et al., 2013). The executive memory head records every sequence of actions that led to pass a level without any kill. Then, when facing the same level, the head gives a high value to the recorded action, in order to force the aggregator's selection. *Nota bene:* since it does not allow generalisation, this head is only employed for the level-passing experiment.

### A.3   A3C baselines

Since we use low level features for the HRA architecture, we implement A3C and evaluate it both on the pixel-based environment and on the low-level features. The implementation is performed in a way to reproduce results of Mnih et al. (2015).

They are both trained similarly as in Mnih et al. (2016) on $8.10^8$ frames, with $\gamma = 0.99$, entropy regularisation of 0.01, $n$-step return of 5, 16 threads, gradient clipping of 40, and $\alpha$ is set to take the maximum performance over the following values: $[0.0001, 0.00025, 0.0005, 0.00075, 0.001]$. The pixel-based environment is a reproduction of the preprocessing and the network, except we only use a history of 2, because our steps are twice as long.

With the low features, five channels of a 40 by 40 map are used embedding the positions of Ms. Pac-Man, the pellets, the ghosts, the blue ghosts, and the special fruit. The input space is therefore 5 by 40 by 40 plus the direction appended after convolutions: 2 of them with 16 (respectfully 32) filters of size 6 by 6 (respectfully 4 by 4) and subsampling of 2 by 2 and ReLU activation (for both).

Figure 3: Gridsearch on $\gamma$ values without executive memory smoothed over 500 episodes.

Figure 4: Gridsearch on $\gamma$ values with executive memory.

Then, the network uses a hidden layer of 256 fully connected units with ReLU activation. Finally, the policy head has $nb_{actions} = 9$ fully connected unit with softmax activation, and the value head has 1 unit with a linear activation. All weights are uniformly initialised He et al. (2015).

### A.4 Results

**Training curves.** Most of the results are already presented in the main document. For more completeness, we propose here the results of the gridsearch over $\gamma$ values for both with and without the executive memory. Values $[0.95, 0.97, 0.99]$ have been tried independently for $\gamma_{score}$ and $\gamma_{ghosts}$.

Figure 3 compares the training curves without executive memory. We can notice the following:

- all $\gamma$ values turn out to yield very good results,
- those good results generalise over random human starts (not shown),
- high $\gamma$ values for the ghosts tend to be better,
- the $\gamma$ value for the score is less impactful.

Figure 4 compares the training curves with executive memory. We can notice the following:

- the comments on Figure 3 are still holding,
- it looks like that there is a bit more randomness in the level passing efficiency.