[Reviews · NeurIPS 2017]

Reviewer 1



R5: Summary: This paper builds on the basic idea of the Horde architecture: learning many value functions in parallel with off-policy reinforcement learning. This paper shows that learning many value functions in parallel improves the performance on a single main task. The novelty here lies in a particular strategy for generating many different reward functions and how to combine them to generate behavior. The results show large improvements in performance in an illustrative grid world and Miss Pac-man. Decision: This paper is difficult to access. In terms of positives: (1) the paper lends another piece of evidence to support the hypothesis that learning many things is key to learning in complex domains. (2) The paper is well written, highly polished with excellent figures, graphs and results. (3) The key ideas are clearly illustrated in a small domain and then scaled up to Atari. (4) The proposed approach achieves substantial improvement over several variants of one of the latest high-performance deep learning systems. My main issue with the paper is that it is not well paced between two related ideas from the literature, namely the UNREAL architecture [1], and Diuk’s [2] object oriented approach successfully used many years ago in pitfall—a domain most Deep RL systems perform very poorly on. It’s unclear to me if these two systems should be used as baselines for evaluating hydra. However, it is clear that a discussion should be added of where the Hydra architecture fits between these two philosophical extremes. I recommend accept, because the positives outweigh my concerns, and I am interested to here the author’s thoughts. The two main ideas to discuss are auxiliary tasks and object-oriented RL. Auxiliary tasks are also built on the Horde idea, where the demons might be predicting or controlling pixels and features. Although pixel control works best, it is perhaps too specific to visual domains, but the feature control idea is quiet general. The nice thing about the UNREAL architecture [1] is that it was perhaps one of the first to show that learning many value functions in parallel can improve performance on some main task. The additional tasks somehow regularize the internal state resulting in a better overall representation. The behavior policy did not require weighting over the different value functions it simply shared the representation. This is a major point of departure from Hydra; what is the best design choice here? Much discussion is needed. Another natural question is how well does pixel control work in Miss Pac-man, this was not reported in [1]. I see hydra as somewhat less general than auxiliary tasks idea. In Hydra a person must designate in the image what pixel groupings correspond to interesting value functions to learn. One could view Hydra as the outcome of UREAL pixel control combined with some automatic way to determine which pixel control GVFs are most useful for sculpting the representation for good main task behavior. In this way Hydra represents a benchmark for automatic GVF construction and curation. On the other extreme with have Diuk’s [2] object oriented approach in pitfall. Assuming some way to extract the objects in an Pitfall screen, the objects were converted into a first-order logic representation of the game. Using this approach, optimal behavior could be achieved in one episode. This is interesting because SOTA deep-learning systems don't even report results on this domain. Hydra is somewhat like this but is perhaps more general. Carefully and thoughtfully relating the Hydra approach to these extremes is critical for accurately discussing the precise contribution of hydra and this paper under review here, and I invite the authors to do that. The final result of the paper (honestly admitted by the authors) exploits the determinist of Atari domain to achieve super-human play in miss packman. This result shows that A3C and others do not exploit this determinism. I am not sure how generally applicable this result is. Another method not discussed, Model Free Episodic Control [3], performs well in Miss Pac-man—much better than A3C. Episodic control defines a kernel function and memory built from agent states generated by a deep network. This allows the system to quickly the query the values of related previously seen states, resulting in much faster initial learning in many domains. I would be interested to see the performance of episodic control in miss pac-man with both representations. It is interesting that DQN does not seem to exploit the additional domain knowledge in the fruit experiment. I am trying to figure out the main point here. Is the suggestion that we will usually have access to such information, and hydra will often exploit it more effectively? Or that prior information will usually be of this particular form that hydra uses so effectively? Could unreal’s aux tasks make use of manually ignoring irrelevant features? If that is not feasible in UNREAL, this would be yet another major difference between the architectures! This raises an interesting question: how often can this feature irrelevance be automatically observed and exploited without causing major problems in other domains. It seems like UNREAL’s independent GVFs could eventually learn to ignore parts of it’s shared representation. Low-dim representations: this is a meme that runs throughout the paper, and a claim I am not sure I agree with. Several recent papers have suggested that the capacity of deep learning systems is massive and that they can effetely memorize large data-sets. Do you have evidence to support this seemingly opposite intuition, outside of the fact that Hydra works well? This seems like a post-hoc explanation for why hydra works well. This should be discussed more in the paper [1] Jaderberg, M., Mnih, V., Czarnecki, W. M., Schaul, T., Leibo, J. Z., Silver, D., & Kavukcuoglu, K. (2016). Reinforcement learning with unsupervised auxiliary tasks. arXiv preprint arXiv:1611.05397. [2] Diuk, C., Cohen, A., & Littman, M. L. (2008, July). An object-oriented representation for efficient reinforcement learning. In Proceedings of the 25th international conference on Machine learning (pp. 240-247). ACM. [3] Blundell, C., Uria, B., Pritzel, A., Li, Y., Ruderman, A., Leibo, J. Z., ... & Hassabis, D. (2016). Model-free episodic control. arXiv preprint arXiv:1606.04460. Small things that did not effect the final decision: line 23: performed > > achieved 26: DQN carries out a strong generalisation > > DQN utilizes global generalization—like any neural net— 28: model of the optimal value function. I don’t think you mean that 230: how important are the details of these choices? Could it be -500 or -2000?


Reviewer 2



This paper proposes splitting up the Q-value prediction output of a DQN net into multiple different GVFs. Each GVF is associated to a particular aspect of the task at hand (this is defined by the user) and these GVFs are then combined in some manner to predict the expected return (aka actual Q-value). Performance is then demonstrated on two tasks to be state of the art. Opinion: To begin with, I am not entirely sure how this differs from the "REINFORCEMENT LEARNING WITH UNSUPERVISED AUXILIARY TASKS" paper. The network architecture is not identical, but the fact that you are seeding your meta-agent with a bunch of local prediction tasks to speed up and robustify the overall Q-value prediction seems to be the same. That aside, here are some comments specific to your paper: 1. I am confused as to how you are aggregating your individual Q-values into the final Q-value. From the definition of Q_hydra (Eq 7.5 (since there is no actual equation label)), it is a linear combination of the individual GVFs for the sub-policies, so I would assume you would aggregate according to the weights w_i. However, on line 121 you mention a max or mean-based aggregation. Indeed, if you use an arbitrary aggregation technique you are unlikely to find Q^*_env, but I am quite confused as to why you would use any other aggregation other than learned linear combinations of your GFVs. 2. It's unfortunate that the tasks need to be specified by the user, and can't be somehow automatically determined. This requires significant task-specific engineering and is not really in the spirit of making general control algorithms that can be easily applied to arbitrary problems. I am also generally confused by your choice of high-level domain knowledge elements that you could leverage to improve performance. For point 1, is the pruning to be done in an automated way? (L1 regularization on the weights of the final value function comes to mind, but doesn't seem to be the chosen approach). For points 2 I am not sure how explicitly predicting the presence of a terminal state would 'free up' weights to better predict values elsewhere. For point 3 it seems that pseudo-rewards is a superset of your reward-function decompositions, but to be honest, until I started getting confused by this point, I assumed you were using arbitrary features of the environment, and not necessarily limitting yourself to a-priori known components of the reward function. 3. Your state of the art is missing many elements, notably the unsupervised auxiliary task paper, but also the "Universal Value Function Approximator" paper by Schaul (and I'm sure quite a bit more, as both those papers have significantly larger related work sections). I'm not judging by the count of papers, but the reduced section makes it difficult to properly compare and contrast your approach to existing and very similar approaches. 4. I am not entirely satisfied with the experimental results. I have a series of concerns with the Ms. Pacman results: 4.1 -- The choice of aggregator types seems somewhat arbitrary, is there any particular theoretical or intuitive motivation for the two aggregation methods? 4.2 -- You introduce very briefly your executive-memory node. Would this node function in a non-deterministic environment? Did you try running without this node? It seems to me almost all your performance benefits may be coming from this one aspect, which is very specific to a deterministic environment. Overall, the experimental section seems to rely on a series of very task-specific design decisions. It does not leave me enough clarity if the high scores on Ms. Pacman have anything to do with your particular approach to reward decomposition, or if the addition of these 3-4 task-specific design choices are responible. My final opinion is that this paper introduces a series of interesting ideas, but in a hard to follow and somewhat unstructured manner. I am not satisfied with how you motivate the reward decomposition as well as the manner in which the rewards are aggregated. I am also not satisfied that the performance on Ms. Pacman has anything to do with your reward decomposition and not with your executive-memory trick. I understand that objectively comparing an algorithm which is designed to exploit domain knowledge with a series of more general algorithms is tricky, but if you want to exploit domain knowledge please use more real-world tasks that are hard to design algorithms for in general. Overall, although I appreciate the effort that went into this paper, I don't believe it is clear, concise and rigorous enough for inclusion into NIPS proceedings. Details: Line 97: Q-value functions are not typicall approximated with a deep net, this is still a relatively new occurence. The reason you are likely using a deep convolutional net in this paper is because you are using 2d images as input and these are currently best processed with neural nets. Line 172 I believe you meant HYDRA+1 and so on (and not HYDRO). Line 205: "with without"

Reviewer 3



This paper presents a novel way of decomposing the reward function into multiple smoother reward functions, thus making it easier to learn the task. In addition, decomposing the reward function this way enables things like doing state abstraction per new reward function, and specifying other domain info per reward signal. The authors demonstrate this learning speedup on a toy domain as well as the Atari game Ms PacMan. And with some additional engineering, they achieve maximum possible scores on Ms Pac-Man (super-human scores). De-composing the reward function into multiple smoother rewards makes a ton of sense, and fits the existing Horde architectures very nicely. It is also really nice that it enables extra domain knowledge to be easily integrated in. One of the main benefits of this approach is the ability to easily add domain knowledge. This is very much emphasized in the Atari results and conclusions, but it is never mentioned in the intro or abstract - it should be included there as well. It's very clear which domain knowledge is being injected into HYDRA in the toy domain (i.e. HYDRA-1, HYDRA-2, etc), but it's not clear for the Atari case. Which pieces of domain knowledge is it using? All of them? The extra head that repeats sequences of actions that beat a particular level is very hand engineered for Ms Pacman and the particular deterministic eval metric being used. It results in tremendous scores on Ms Pacman, but it feels like a hack added on just for those scores. I don't think its related to the contribution of the paper in any way (I imagine you could add this memory replay head to any agent and eventually defeat Ms Pacman this way). Some technical questions: In equation 6, would it not make sense to weight the inidividual reward Q losses by the weights being used in the Q_hydra aggregation? In equation 7, instead of doing the max over that particular reward's Q-function, you could use Q_hydra for the argmax and then that individual rewards Q-function for the target values. Then all of the Q-functions would be consistently updating with the same max action and I think Q*_hydra should then equal Q*_env. Small edits: Line 172 says HYDRO instead of HYDRA Line 222: starts of -> starts off